# Low Surface Roughness Graphene Oxide Film Reduced with Aluminum Film Deposited by Magnetron Sputtering

**DOI:** 10.3390/nano11061428

**Published:** 2021-05-28

**Authors:** Xiaowei Fan, Xuguo Huai, Jie Wang, Li-Chao Jing, Tao Wang, Juncheng Liu, Hong-Zhang Geng

**Affiliations:** 1Tianjin Key Laboratory of Advanced Fibers and Energy Storage, School of Material Science and Engineering, Tiangong University, Tianjin 300387, China; xiaowei_fan@126.com (X.F.); wangjie01@visionox.com (J.W.); jinglctjpu@163.com (L.-C.J.); wangtao145411@foxmail.com (T.W.); jchliu@tiangong.edu.cn (J.L.); 2Center for Engineering Internship and Training, Tiangong University, Tianjin 300387, China

**Keywords:** graphene oxide, nascent hydrogen, reduction, aluminum film, magnetron sputtering

## Abstract

Graphene film has wide applications in optoelectronic and photovoltaic devices. A novel and facile method was reported for the reduction of graphene oxide (GO) film by electron transfer and nascent hydrogen produced between aluminum (Al) film deposited by magnetron sputtering and hydrochloric acid (HCl) solution for only 5 min, significantly shorter than by other chemical reduction methods. The thickness of Al film was controlled utilizing a metal detection sensor. The effect of the thickness of Al film and the concentration of HCl solution during the reduction was explored. The optimal thickness of Al film was obtained by UV-Vis spectroscopy and electrical conductivity measurement of reduced GO film. Atomic force microscope images could show the continuous film clearly, which resulted from the overlap of GO flakes, the film had a relatively flat surface morphology, and the surface roughness reduced from 7.68 to 3.13 nm after the Al reduction. The film sheet resistance can be obviously reduced, and it reached 9.38 kΩ/sq with a high transmittance of 80% (at 550 nm). The mechanism of the GO film reduction by electron transfer and nascent hydrogen during the procedure was also proposed and analyzed.

## 1. Introduction

Graphene, a rapidly rising star for industry, which is arranged in a hexagonal lattice with sp^2^ hybridized carbon atoms owning a peculiar two-dimensional crystal structure, is the thinnest membrane at present [1,2]. Due to the high optical transparency and electrical conductivity [3], transparent conductive films (TCFs) made by graphene have been used in different photovoltaic and optoelectronic devices, such as transparent flexible heaters [4], organic lighting devices [5], flexible touch screens [6], solar cells [7], and photosensors [8].

In recent years, graphene oxide (GO) has been regarded as the important precursor to fabricate transparent conductors [9]. Depositing graphene oxide sheets on a flexible substrate to fabricate TCFs has been studied by many researchers in the field. Since GO dispersions are suitable to produce films on any substrate [10], TCFs composed of GO or reduced GO sheets (rGO) have been prepared via many techniques, including spray coating [11], transfer printing [12], spin coating [13], electrophoretic deposition [14], rod coating [6], and the Langmuir–Blodgett assembly [15], etc. Among these methods, the rod coating method has been widely used in making solution-processed thin films for mass production in the coating industry.

GO is insulating because the sp^2^ structure is broken [9]. However, the conductivity of the structure can be recovered via fixing the π–π network [16], and the reduction of GO is the key to restoring the excellent conductivities of graphene. Thermal and chemical reduction are the two main methods to reduce graphene oxide [17,18]. The high-temperature heat treatment method usually requires at least 1000 °C, which requires that the film substrates are capable of withstanding high temperatures, thus limiting its wide application. The chemical reduction methods are generally highly toxic and dangerous due to the use of chemicals such as hydrazine [19], dimethylhydrazine [20], and sodium borohydride [21], causing serious environmental pollution if heavily utilized. The above chemicals are not proper to reduce GO films, which need high flexibility for the applications of flexible devices.

The GO film currently used for the fabrication of TCFs is inaccessible in industry because of difficulties in its mass production and reduction. Using a metal and acid mixture for the reduction of GO has gained more and more attention because of its very fast and efficient reduction abilities [22,23]. Moreover, as is well known, rod coating is conducted for roll-to-roll mass production in industry [24]. Therefore, an effective and environment-benign reduction method for the chemical reduction of GO film is the key to the preparation of TCFs for commercial applications. It is well known that metal deposition by magnetron sputtering is widely used for coating in various industrial fields. This technology has the advantage of being able to coat a uniform and dense Al film on any substrate. Moreover, Al can be partially deposited by shielding or shadow-masking, which can generate well-designed patterns. Therefore, combining the rod-coating method with the magnetron sputtering technology, we will achieve the large-size production of graphene-based TCFs for industry.

This work presents a facile and rapid route to the metal-mediated reduction of GO film by electron transfer and nascent hydrogen produced between Al film deposited by the magnetron sputtering method and hydrochloric acid (HCl) solution for the first time. In this experiment, the GO films were produced on a poly(ethylene terephthalate) (PET) substrate utilizing the Meyer rod coating method [6]. Al films can be formed by magnetron sputtering. In the gaseous state, Al atoms attached onto the GO film, and then Al atoms cooled and crystallized to form Al-GO film. Then, the Al-GO films were immersed in HCl solution and reacted with HCl to create Al-reduced GO (Al-rGO) film after only 5 min at room temperature. The effects of the deposited Al film thickness and the HCl solution concentration during the reduction process will be further analyzed. After reduction, the sheet resistance of the Al-rGO film decreased from 22.39 to 6.08 kΩ/sq and the transmittance dropped from 87.0% to 68.2%. Furthermore, atomic force microscope (AFM) images indicated that the roughness of the Al-rGO film was significantly decreased. Though the sheet resistance is still higher for application, with further optimization, large-area GO film can be produced through the rod coating method and Al-deposited GO film via the industry magnetron sputtering method, which would allow the fabrication of large-area and low-cost Al-rGO films on any kind of substrate—the processes do not need any heating and transferring technology. The mechanism of the reduction of GO film by electron transfer and nascent hydrogen during the procedure was also analyzed.

## 2. Materials and Methods

### 2.1. Preparation of GO Suspension

GO was prepared by the modified Hummers method using natural graphite flakes [25]. Briefly, 2 g flake graphite and 1.5 g NaNO_3_ (95%) were mixed in concentrated H_2_SO_4_ (98%), and 9 g KMnO_4_ (99.5%, Sinopharm Chemical Reagent Co., Ltd., Shanghai, China) was carefully added during vigorous stirring. The stirring was continued for 5 d. Then, 200 mL deionized water (DI water) was added into the mixture, and then 10 mL 30% H_2_O_2_ was added to complete the oxidation. The obtained mixture was rinsed using 5% HCl solution and DI water by centrifugation again and again until PH = 7. Lastly, the GO suspension was gained by ultrasonication using a bath sonicator, and the suspension then underwent mild centrifugation to obtain a uniform GO solution.

### 2.2. Preparation of Al-rGO Film

The experimental procedure is shown in Scheme 1. Firstly, the GO suspension was coated onto PET substrate using a Meyer rod to obtain GO film (Scheme 1(a,1) corresponding GO structure [26]). The GO film thickness was controlled by the wire diameter and the GO solution concentration. Al-GO films were prepared by an industry magnetron sputtering system (VA-3348, Von Arddenne, Dresden, Germany) under Ar atmosphere. The Al material was selected as the sputtering target. The vacuum inside the sputtering chamber was ~5 × 10^−4^ pa. The thicknesses of Al films were monitored using a digital thickness monitor connected with a dual quartz crystal microbalance inside the sputtering chamber. Al atoms were sputtered onto GO films to form uniform and dense Al-GO films, and the different thicknesses of GO films were controlled by the sputtering time and power. After achieving the desired thickness of Al-material, the deposition process was stopped (Scheme 1(b,2) corresponding GO structure with Al atoms). Finally, Al-GO films were immersed in HCl solution (10 M) for 5 min, and subsequently washed with dilute HCl solution and DI water several times. Al-rGO films were obtained after drying at 80 °C in the oven (Scheme 1(d,3) corresponding rGO structure).

### 2.3. Characterization

The surface morphology and roughness of GO/Al-rGO films were recorded with AFM (CSPM5500, Being Nano-Instruments, Guangzhou, China) and a field-emission scanning electron microscope (FE-SEM, Hitachi S-4800, Tokyo, Japan). The thicknesses of the GO films and Al films deposited on quartz substrates were characterized by AFM. The reduction of GO film was characterized using X-ray photoelectron spectroscopy (XPS, ELMER PHI 5600, PerkinElmer, MA, USA). The sheet resistance (*R*_s_) was measured using a four-point probe (Keithley 2400, Tektronix, Beaverton, OR, USA). The transparency of GO and Al-rGO films was measured using UV-Vis spectroscopy (UV-Vis spectrometer, Lambda 35, PerkinElmer). Fourier transform infrared spectroscopy (FT-IR, TENSOR 37, Bruker, Billerica, MA, USA), X-ray diffraction (XRD, D/MAX-2500, Rigaku, Tokyo, Japan), high-resolution transmission electron microscopy (HR-TEM, Tecnai G2 F20, FEI, Hillsboro, OR, USA), and Raman spectra (Thermo Scientific DXR, Waltham, MA, USA) were also carried out.

## 3. Results

The reduction degree of the GO films with different Al thicknesses was characterized using UV-Vis spectroscopy. The GO film’s absorption peak was 230 nm [27], the UV-Vis absorption peak red-shifted, and the absorbance in the whole spectral region was also increased due to the reduction [28], as shown in Figure 1a. Using the GO film of 68% transparency (thickness of 10 nm, Figure 2a) as an example selected from Figure 3b, the absorption peak of the original GO film was located at 230 nm, and then it gradually red-shifted to 268 nm along with the increase in the thickness of Al film (Figure 2b–d) for the reduction. This red-shift suggested that part of the conjugated C=C bonds were restored. The absorption peak red-shifted to 268 nm when the Al thickness increased to 15 nm and remained the same even if the Al thickness rose. The curve of the position of the absorption peak as a function of the thickness of Al film is shown in Figure 1b. The absorption peak around 230 nm red-shifted towards 268 nm and the surface resistance decreased gradually with the increase in the Al film thickness. The results indicate the restoration of the sp^2^-bonded carbon network and the improvement of the electrical conductivity.

The height profiles of GO film and Al films were characterized by AFM (shown in Figure 2). The thickness of the GO film with 68% transmittance after reduction was 10 nm (Figure 2a), and the thicknesses of Al films are 10, 15, and 20 nm, respectively (Figure 2b–d). The profiles of AFM images (Figure 2b–d) of Al films are very smooth, which indicates the uniform deposition of Al using magnetron sputtering. Combined with the sheet resistance changing with the Al thickness, the optimal thickness of the Al film was chosen to be 15 nm for the 10-nm GO film, so that Al-rGO film can reach its best performance without wasting more Al. Since Al atoms were deposited uniformly during the sputtering process, the morphology of Al film barely influences the reduction of GO films.

The effect of the HCl concentration with the sheet resistance of Al-rGO film was investigated, as shown in Figure 3a. The 10 M concentration gives the best performance, displaying the lowest sheet resistance. When the HCl concentration is too high, the reducing agent Al cannot efficiently react with GO. The corresponding curve of the *R_s_* versus the transmittance of Al-rGO films treated with 10 M HCl solution is shown in Figure 3b. When the transmittance (at 550 nm) increases from 68.2% to 87.0%, the sheet resistance varies from 6.07 to 22.39 kΩ/sq. The as-prepared Al-rGO film exhibits a *R*_s_ of 9.38 kΩ/sq at the transmittance of 80% (550 nm). The *R*_s_ and transmittance of Al-rGO film can be easily controlled by changing the thicknesses of GO and Al films. The property of the rGO film not only depends on the reduction method, but also depends on the quality of GO sheets and GO film. Comparing with rGO films fabricated by other processes, the Al-rGO film obtained by this method displays outstanding properties, with high transparency and good conductivity (shown in Appendix A).

To further characterize the effect of the reduction of GO film, the elemental analysis for the GO film and Al-rGO film was performed by XPS. The C1s and O1s peaks of GO and Al-rGO film can be observed in the whole spectra, as shown in Figure 4a. The analysis of the XPS spectra indicates that the C/O ratios increase from 1.4 to 9.5. The reduction effect is obvious because the intensity of the O1s band of Al-rGO films extensively decreases compared with GO film since a number of oxygen-containing functional groups were removed from GO film [29]. Furthermore, there is no peak (at ~73 eV) for the Al element in Al-rGO film, meaning no Al residual after reduction. It also shows that the Al used for reduction has no doping effect in the process. Additionally, based on the C1s XPS spectrum of GO (Figure 4b), the deconvolution of the C1s peak of GO exhibits five peaks, which are located at the binding energies of 284.5, 285.5, 286.8, 287.8, and 288.9 eV, and can be assigned to the C–H (C=C, C–C), C–OH, epoxide, C=O (carbonyl C), and COOH (carboxylate C) functional groups, respectively [30,31,32]. Though the C1s XPS spectrum of the Al-rGO film (Figure 4c) also exhibits these oxygen-containing functional groups, the intensity of most peaks is much weaker than those in the GO film (Figure 4b). The contents (at.%) of oxygen-containing functional groups (sp^2^C, sp^3^C, C–O, –C=O, and –COO–) of the GO film are 33.6, 18.0, 36.0, 7.5, and 4.9, respectively, and those of Al-rGO become 67.3, 14.6, 7.2, 6.0, and 4.9, respectively. The oxygen-containing functional groups decrease extensively, especially for the peak of C–O (epoxy and alkoxy), disclosing that almost all the oxygen-containing functional groups were reduced.

To better illustrate the ability of Al to reduce GO, FT-IR, Raman, and XRD measurements were conducted on both GO and Al-rGO samples. The reduction of the oxygen-containing groups can be confirmed via comparing the FT-IR spectra of the GO and Al-rGO samples (Figure 5a). The spectra of GO at 3430 cm^−1^ are attributed to the O–H stretching vibrations. The spectrum of GO also reveals the existence of the C=O stretching vibration peak at 1725 cm^−1^, the C–O–C (epoxy) stretching vibration peak at 1221 cm^−1^, and the C–O (alkoxy) stretching peak at 1050 cm^−1^, respectively. The intensities of the FT-IR peaks for Al-rGO film decreased dramatically after the reduction, and some even disappeared [33,34].

The Raman spectra of GO and Al-rGO samples are characterized by two main features (Figure 5b). The G-band at 1580 cm^−1^ results from the in-plane bond stretching of the sp^2^ carbons. The D-band located at 1350 cm^−1^ can be assigned to the defect and the size of the in-plane sp^2^ domain. It is well known that the ratio between the peak intensities of D- and G-bands is an indicator of the reduction degree. The intensity ratio of the D- to G-band (*I*_D_/*I*_G_) is enhanced from 0.98 to 1.16. The possible explanation is that the reduction enhances the amount of aromatic domains of slighter overall size in graphene, leading to the increment of the *I*_D_/*I*_G_ ratio [29]. To determine the decrease in the average size of the sp^2^ domain, the Tuinstra and Koenig relation was employed, which transmits the ratio of D- to G-band into the crystallite size [10]:*I*_D_/*I*_G_ = *C*(*λ*)/*L*_a_(1)
where *I*_D_/*I*_G_ is the intensity ratio of the D- to G-band, *L*_a_ is the average crystallite size of sp^2^ domains, and *C*(*λ*) is the wavelength-dependent prefactor. The increased ratio of *I*_D_/*I*_G_ results in the *L*_a_ decrease, owing to the creation of a new sp^2^ domain which is smaller in the range compared to that which appeared in GO before reduction [35].

Figure 5c shows the XRD results from graphite, GO, and Al-rGO samples. Due to the oxidation of pristine graphite, the XRD of GO exhibits a (001) diffraction peak (2θ = 13°) with an interlayer space of 0.59 nm. After reduction, the peak moved backwards to 2θ = 22.92°, corresponding to a d spacing of 0.388 nm, compared with that of pristine graphite at 26.6°, indicating that most of the oxygen-containing functional groups were eliminated by reduction. The conjugated graphene network was restored by the reduction [36,37].

Magnetron-sputtered Al film can reduce GO film to Al-rGO film. The photographs of GO film and Al-rGO film are shown in Figure 6a,b, in which the appearance of Al-rGO film is almost unchanged compared with the GO film. Both GO film and Al-rGO film consist of overlapping platelets and keep the sheet morphology [38] (Figure 6c,d). It can be observed that the color changed from light to dark brown during the reduction of GO film, and the transparency became a little worse [39].

It can also be observed in AFM (Figure 7a,b) that there are extensive networks of long, broad wrinkles across the film surface on both GO film and Al-rGO film [13]. Although the GO film is rough, the surface roughness (*R*_q_) of the reduced GO film is significantly declined. The *R*_q_ is decreased from 7.68 to 3.13 nm. To compare with other reduction methods, a comparison of several different reducing agents toward the *R*_q_ of GO film, such as hydroiodic acid (HI) [40] and hydrazine [41], was carried out. As can be seen from Figure 6c,d, the *R*_q_ of HI-reduced GO film (HI-rGO) and hydrazine-reduced GO film (hydrazine-rGO) are 5.75 and 6.22 nm, respectively. These chemical reagents are strong reducing agents, which cannot effectively decrease the *R*_q_ of the GO films. Thus, the method using Al as a reducing agent can reduce the *R*_q_ and maintain the morphology of the GO films, suggesting that it is a relatively modest reduction method to reduce GO films with very low surface roughness.

The reduction of GO film is a process to remove the oxygen-containing functional groups (epoxy, carbonyl, and hydroxyl groups) [28]. For GO reduction by Al, GO can be converted to graphene (G) in the presence of Al and HCl solution within a few minutes. The probable reduction mechanism is shown in Scheme 2. On one hand, the GO was directly reduced by the electron transportation [25,28]. It is known that Al is an active metal and has huge reducing ability, similar to Zn [23], Fe [25], as well as other active metals for GO reduction in the presence of electron transport from the metal to the GO film. The electron transfer reaction can be concluded in the following equations ①:Al → Al^3+^ + 3e^−^(2)
GO + nH^+^ + ne^−^  → G + mH_2_O(3)

Al film is closely covered on the surface of GO film, which facilitates the reduction of GO film owing to the rapid electron transfer from Al to GO film. On the other hand, the H atom is a capable reducing agent for reducing GO film. The promising reducing mediator for the reduction of GO film is nascent hydrogen [28,42,43], denoted as H·, which is produced from the reaction of Al and HCl solution. The reduction of GO film with nascent hydrogen is shown below ②:Al → Al^3+^ + 3e^−^(4)
H^+^ + e^−^ → H·(5)
GO + 2H· → G + H_2_O, Al → Al^3+^ + 3e^−^(6)

Nascent hydrogen is a strong reducing agent which can willingly react with carbonyl, epoxy, and hydroxyl groups. When Al is exposed to H^+^ in HCl solution, active H atoms generated can efficiently reduce GO film to Al-rGO film, and this leads to the recovery of the conjugated C=C bonds in the graphene basal plane. Scheme 2 shows the proposed reaction mechanism.

Pham et al. [28] suggested that the nascent hydrogen can be employed to reduce GO effectively. However, just a few GO layers on the GO film surface can be reduced in a short time. Although Al seems to be deposited only on the surface of GO film, the electrons are transported through the reduced GO sheets, which are in contact with Al as an electron mediator to the layers underneath the surface for reduction [31]. The electrons generated during the reaction of Al and HCl can transport faster to the inner layers of GO film than nascent hydrogen. Therefore, we suggest that the reduction with nascent hydrogen mainly occurred on the surface of GO film and the main reduction should be the electron transfer in the whole reduction process.

Since Al atoms are in close contact with GO film, when reacting with HCl solution, the occurrence of electron transfer is accompanied by the production of nascent hydrogen. Therefore, the two reduction mechanisms can be incorporated in the entire reaction in the meantime. In an acidic environment, Al^3+^ is inhibited to ionization; the high H^+^ concentration was beneficial to the reduction reaction for electron transfer and the generation of nascent hydrogen. It also can be explained to a degree that, by raising the HCl concentration, the Al-rGO film’s conductivity increases within limits. However, if the concentration of HCl solution is too high, the electrons and nascent hydrogen between Al and HCl are produced so quickly that the nascent hydrogen combines with hydrogen gas, and more hydrogen bubbles are formed as can be detected on the Al-GO film surface during the experimental process, which means a large amount of electrons and nascent hydrogen cannot be efficiently reacted with GO, resulting in an increase in the surface resistance when the concentration of HCl is 12 M, as shown in Figure 3a. Thus, a suitable concentration of HCl solution in this system was beneficial during the reduction. Though the property of Al-rGO film may not be good enough to be used in various applications, with further optimization, large-area TCFs can be produced through the rod coating method combined with Al deposited via the industry magnetron sputtering method, which would permit the manufacture of graphene films on any kind of substrate, exclusive of any toxicant as well as transferring processes.

## 4. Conclusions

This paper proposes a novel and rapid reduction strategy for fabricating flexible Al-rGO films using magnetron sputtering of Al as the reducing mediator in acidic conditions. GO can be reduced by the electron transfer and nascent hydrogen only for 5 min, extensively shorter than by other chemical reduction procedures. UV-Vis absorption spectroscopy shows a red-shift of the absorption peak along with further reduction. For a 10 nm thickness GO film, the optimal Al thickness for the reduction was 15 nm. Further characterization with Raman, FT-IR, and XRD confirmed the effect of the reduction of GO via Al. The film sheet resistance was significantly reduced with Al film by magnetron sputtering and reached 9.38 kΩ/sq while maintaining a high transmittance of 80% at 550 nm. The *R_q_* decreased from 7.68 to 3.13 nm after reduction, showing a very low surface roughness. Even though the optical and electrical properties of these Al-rGO films prepared by this method still require extra enhancement, our work offers a facile and rapid technology for the mass production of graphene-based transparent conducting films.

## Data Availability

Not applicable.

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
