# Peer review of "Low Surface Roughness Graphene Oxide Film Reduced with Aluminum Film Deposited by Magnetron Sputtering"

_nanomaterials, 2021, doi:10.3390/nano11061428_

Round 1

Reviewer 1 Report

The current manuscript by Xiaowei et al., is an interesting study discussing a method for the reduction of graphene oxide film using Al film deposited by magnetron sputtering which the authors also claim it to be a faster approach compared to other known chemical reduction methods.

The manuscript is clearly written and the methods are clearly presented. The quality of figures is very good. GO/Al-rGO films are well characterized using various characterization techniques like AFM, XPS, XRD, UV-Vis spectroscopy, TEM, Raman and FTIR.

The current work is interesting for those working in the field and I recommend the article to be accepted for publication in its present form.

Author Response

We appreciate the reviewer for reviewing our manuscript and regarding that ‘The manuscript is clearly written and the methods are clearly presented. The quality of figures is very good.our paper is generally well-written. The current work is interesting for those working in the field and I recommend the article to be accepted for publication in its present form.’.

Reviewer 2 Report

The paper presents a facile and rapid technology for scalable fabrication of graphene-based transparent conductive films by using of a novel reduction strategy of GO through magnetron sputtering of Al  and hydrochloric acid (HCl) solution. The paper is generally well-written. There are some issues, which need to be clarified.

                Line 110: “The thicknesses of Al films were monitored using a digital thickness monitor”

Can you give more details about the thickness measurement.

                Lines 134-137: “Using the GO film of 68% transparency (thickness of 10 nm, Figure 2a) as an example, the absorption peak of the GO film without reduction appeared at 230 nm, but gradually red-shifted to 268 nm along with the increase of the thickness of Al film (Figure 2b-d) for the reduction.”

This is very unclear. Fig.2 shows AFM images and it is difficult to relate its content with transparency. It will be good to present the transparency of films measured by UV-Vis spectroscopy.

                Lines 154-156: “Therefore, the optimal thickness of the Al film is 15 nm for the 10-nm GO film, so that Al-rGO film can reach its best performance without wasting more Al.”

This conclusion is not very well substantiated.

                In the Conclusion it is written:

“The sheet resistance of the films was significantly reduced with Al film by vacuum evaporation and reached 9.38 kΩ/sq with high transmittance of 80% at 550 nm.”

Is there a mistake with the deposition method of Al? Isn't it prepared by sputtering?

  • English should be carefully checked for grammar and typing errors. Below are several examples:
  • Line 172: The property of the rGO film dependents not only on the reduction method,….;
  • Line 219: The D-band locates at 1350 cm-1 can be assigned…;
  • Line 230: …new sp2 domain which are smaller…;
  • Line 316: Though the property of Al-rGO film may not good enough to be used…
